# Plasma Vitamin B_12_ and Folate Alter the Association of Blood Lead and Cadmium and Total Urinary Arsenic Levels with Chronic Kidney Disease in a Taiwanese Population

**DOI:** 10.3390/nu13113841

**Published:** 2021-10-28

**Authors:** Yu-Mei Hsueh, Ya-Li Huang, Yuh-Feng Lin, Horng-Sheng Shiue, Ying-Chin Lin, Hsi-Hsien Chen

**Affiliations:** 1Department of Family Medicine, Wan Fang Hospital, Taipei Medical University, Taipei 110, Taiwan; ymhsueh@tmu.edu.tw (Y.-M.H.); green1990@tmu.edu.tw (Y.-C.L.); 2Department of Public Health, School of Medicine, College of Medicine, Taipei Medical University, Taipei 110, Taiwan; ylhuang@tmu.edu.tw; 3Graduate Institute of Clinical Medicine, College of Medicine, Taipei Medical University, Taipei 110, Taiwan; linyfmd@tmu.edu.tw; 4Division of Nephrology, Department of Internal Medicine, Shuang Ho Hospital, Taipei Medical University, New Taipei City 235, Taiwan; 5Department of Chinese Medicine, Chang Gung University College of Medicine, Taoyuan 333, Taiwan; hongseng@ms1.hinet.net; 6Department of Family Medicine, School of Medicine, College of Medicine, Taipei Medical University, Taipei 110, Taiwan; 7Department of Geriatric Medicine, School of Medicine, College of Medicine, Taipei Medical University, Taipei 110, Taiwan; 8Division of Nephrology, Department of Internal Medicine, School of Medicine, College of Medicine, Taipei Medical University, Taipei 110, Taiwan; 9Division of Nephrology, Department of Internal Medicine, Taipei Medical University Hospital, Taipei 110, Taiwan

**Keywords:** vitamin B_12_, folate, cadmium, lead, arsenic, chronic kidney disease

## Abstract

Heavy metals causing chronic nephrotoxicity may play a key role in the pathogenesis of chronic kidney disease (CKD). This study hypothesized that plasma folate and vitamin B_12_ would modify the association of CKD with total urinary arsenic and blood lead and cadmium levels. We recruited 220 patients with CKD who had an estimated glomerular filtration rate of <60 mL/min/1.73 m^2^ for ≥3 consecutive months and 438 sex- and age-matched controls. We performed inductively coupled plasma mass spectrometry to measure blood cadmium and lead levels. The urinary arsenic level was determined using a high-performance liquid chromatography–hydride generator–atomic absorption spectrometry. Plasma vitamin B_12_ and folate levels were measured through the SimulTRAC-SNB radioassay. Compared with patients with plasma vitamin B_12_ ≤ 6.27 pg/mL, the odds ratio (OR) and 95% confidence interval of CKD for patients with plasma vitamin B_12_ > 9.54 pg/mL was 2.02 (1.15–3.55). However, no association was observed between plasma folate concentration and CKD. A high level of plasma vitamin B_12_ combined with high levels of blood lead and cadmium level and total urinary arsenic tended to increase the OR of CKD in a dose-response manner, but the interactions were nonsignificant. This is the first study to demonstrate that patients with high plasma vitamin B_12_ level exhibit increased OR of CKD related to high levels of blood cadmium and lead and total urinary arsenic.

## 1. Introduction

Chronic kidney disease (CKD) is characterized by a progressive and irreversible decline in renal function occurring gradually over a period of a few months to many years. In CKD, the kidney gradually loses its ability to filter toxins from blood. Renal impairment in CKD is diagnosed based on a decrease in the estimated glomerular filtration rate (eGFR) to <60 mL/min/1.73 m^2^, the presence of proteinuria, or the presence of pathological abnormalities for at least 3 months [1]. Because of the high prevalence as well as morbidity and mortality of CKD, it has become a global public health concern [2]. In Taiwan, the CKD prevalence was 11.9% in 2008 [3], and the prevalence of end-stage renal disease was the highest in the world from a 2016 report [4]; therefore, CKD is a significant public health issue in Taiwan.

Our recent study reported that the levels of blood lead and cadmium and total urinary arsenic are significantly associated with an increased odds ratio (OR) of CKD, whereas the plasma selenium level significantly reduced the OR of CKD [5]. A Thai study reported that long-term exposure to cadmium and a high urinary cadmium level were associated with a significant decrease in eGFR, resulting in CKD [6]. A Chinese follow-up study showed that the levels of plasma arsenic and lead are associated with a significant annual decline in eGFR after adjustment for demographic variables and risk factors for CKD [7]. Furthermore, an animal study demonstrated that lead causes an inflammatory response, leading to CKD [8]. A high level of lead in the blood was related to proteinuria and eGFR < 60 mL/min/1.73 m^2^ [9]. Exposure to arsenic, lead, and cadmium appears to be related to CKD occurrence [5,6,7,8,9].

Vitamin B_9_ (folate) and vitamin B_12_ are water-soluble vitamins involved in several normal cellar functions. Folate and vitamin B_12_ are vital cofactors in the remethylation pathway in humans [10]. Folate treatment was associated with a decrease in the OR of CKD progression in patients with mild-to-moderate CKD and high B_12_ levels [11]. A review suggested that folate and vitamin B_12_ can be beneficial in CKD treatment [12]. However, the levels of serum folate and vitamin B_12_ were not associated with increased levels of homocysteine and cysteine in patients with CKD and diabetes [13]. Thus, whether plasma folate and vitamin B_12_ prevent CKD remains unclear. Heavy metals with nephrotoxic effects may accumulate gradually and cause CKD [14]. Therefore, this study investigated whether the levels of plasma folate and vitamin B_12_ alter the OR of CKD related to total urinary arsenic and blood lead and cadmium.

## 2. Materials and Methods

### 2.1. Study Participants and Interviews

Eligible participants were nephrology outpatients and adults or elderly people participating in a health examination who had signed an informed consent form and provided blood and urine samples. In total, 220 patients with clinically confirmed CKD and 438 sex- and age-matched controls were recruited from both Taipei Medical University Hospital and Taipei Municipal Wan Fang Hospital between May 2018 and May 2019. All outpatients with CKD received the diagnosis based on biochemical criteria such as blood urea nitrogen, proteinuria, and serum creatinine at the Department of Internal Medicine/Nephrology. Patients with CKD who had an eGFR of <60 mL/min/1.73 m^2^ were diagnosed as having stage 3, 4, or 5 CKD for at least 3 months and did not receive hemodialysis. Those with an eGFR of ≥60 mL/min/1.73 m^2^ were considered healthy controls. The ratio of control participants to patients with CKD was approximately 1.5:1.

We interviewed all study participants and collected their blood and urine samples as described in a previous study [15]. The current study was approved by the Research Ethics Committee of Taipei Medical University, Taiwan (TMU Joint Institutional Review Board N201804024, 25 May 2018–24 May 2019), and was conducted in accordance with the Declaration of Helsinki.

### 2.2. Measurements of Urinary Arsenic Levels

The urinary arsenic level was measured as described previously [16]. The measurement method, detection limits, and standard reference material used served as the quality standard, and samples spiked with a standard solution (recovery rates) are shown in Appendix A. The total urinary arsenic level (μg/g creatinine) was calculated as the sum of the levels of inorganic arsenic (arsenite + arsenate), monomethylarsonic acid, and dimethylarsinic acid after dividing for the level of urinary creatinine, which controls hydration. The measurement of the creatinine level is shown in Appendix A.

### 2.3. Measurements of Blood Lead and Cadmium Levels

Because the literature indicates that the concentration of heavy metals (such as lead or cadmium) in whole blood is a valid marker for long-term exposure [17], this study used red blood cells to measure the concentration of heavy metals. Blood lead and cadmium levels were measured as described previously [18]. The validity and reliability of the measurements and detection limits are listed in Appendix A.

### 2.4. Measurements of Plasma Folate and Vitamin B_12_ Levels

The methods used for measuring plasma vitamin B_12_ and folate levels were described in detail in our recent study [19]. The measurement method, detection limit, and variation coefficient are presented in Appendix A.

### 2.5. Statistical Analysis

Continuous variables are presented as the mean ± standard deviation or median (IQR). We used the Wilcoxon rank sum test to compare differences in continuous variables between patients with CKD and controls. Furthermore, we used the chi-square test to examine the distribution of categorical variables between patients with CKD and controls. We used a multivariate linear regression model to determine the correlation between eGFR and the levels of plasma folate and vitamin B_12_ after adjusting for age; sex; educational level; alcohol, coffee, and tea consumption; analgesic usage; history of diabetes and hypertension; red blood cell lead and cadmium levels; and total urinary arsenic (μg/g creatinine). Subsequently, we used multiple logistic regression to assess the association between potential risk factors for CKD. The corresponding tertiles of controls were used as cutoff points for continuous variables among independent variables. This approach is generally a dose-response analysis method, which analyzes the increased CKD risk when the dose of the exposure variable increases by one-third [20]. Multivariate-adjusted ORs and 95% confidence intervals were calculated to determine CKD risk. In the significance test of the linear trend of the OR in the exposed stratification, we used categorized exposure variables as score variables, which also served as continuous variables. The respective median of controls was used as the cutoff for risk factors in the interaction analysis. Additive interactions between risk factors for CKD were evaluated in a pairwise manner by using the synergy index provided by Rothman [21]. The observed synergy index value was not equal to 1, indicating an additive interaction, and ORs and their variance–covariance matrix were used to calculate 95% confidence intervals [22]. The product term between levels of plasma vitamin B_12_, blood lead and cadmium, and total urinary arsenic was used pairwise to test their multiplicative interactive effect on the OR of CKD in the multiple logistic regression model. The SAS package (version 9.4; SAS Institute, Cary, NC, USA) was used for these analyses. A two-tailed *p* value of <0.05 indicated statistical significance.

## 3. Results

Table 1 lists the sociodemographic characteristics, lifestyle, and disease histories of patients with CKD and controls. CKD cases and controls were not statistically different in age, sex, and smoking status. However, CKD cases were less educated, less likely to consume alcohol, coffee, or tea, but were more likely to use analgesics and were more likely to be diabetic or hypertensive.

We analyzed the relationship of plasma nutrients, blood lead and cadmium, and urinary metals with CKD risk (Table 2). The higher the levels of plasma vitamin B_12_, blood lead and cadmium, and total urinary arsenic, the higher the OR of CKD. When the concentration of blood lead, cadmium, urinary total arsenic, or plasma vitamin B_12_ increased by a tertile, the risk of CKD increased significantly. Plasma folate levels were not related to CKD (Table 2). We also show the spread of data in Appendix A.

The log eGFR decreased significantly with the increase of the log plasma vitamin B_12_ concentration. However, there was no correlation between plasma folate concentration and eGFR (Figure 1).

Because plasma vitamin B_12_ was related to CKD, we conducted a stratified analysis to determine whether it affects the association of blood cadmium and lead or total urinary arsenic concentration with CKD risk. The effect of blood lead concentration on the OR of CKD in patients with a low plasma vitamin B_12_ level was higher than that in patients with a high plasma vitamin B_12_ level. The OR of CKD did not vary between blood cadmium and total urinary arsenic concentrations (Appendix A). Subsequently, we examined the interactive effects of plasma vitamin B_12_, total urinary arsenic, and blood lead and cadmium levels on CKD (Table 3). A trend analysis revealed that the OR of CKD gradually but significantly increased with exposure to no risk factors or to either one or both risk factors (a high plasma vitamin B_12_ level and a high blood lead level). Furthermore, the interaction of other paired risk factors exerted the same effect on CKD; however, these interactions were nonsignificant.

## 4. Discussion

The results of the present study revealed that the prevalence of hypertension and diabetes was higher in patients with CKD than in controls. Hypertension and diabetes are crucial risk factors for CKD [23]. Furthermore, the present study demonstrated that the increase in plasma vitamin B_12_, total urinary arsenic, and blood lead and cadmium levels gradually and significantly increased the OR of CKD. In addition, high levels of plasma vitamin B_12_ and blood lead and cadmium tended to increase the OR of CKD, but the interaction was nonsignificant.

Our study demonstrated a significantly positive correlation of blood cadmium and lead and total urinary arsenic levels with the OR of CKD [5,24]. In addition, this study also found that urinary total arsenic and blood lead and cadmium were related to CKD, as proposed in other studies. One study did not find an association between the blood lead level and kidney function [25]. However, a cohort study found that plasma arsenic was associated with an increased risk of kidney graft failure [26]. A Thai study showed that long-term exposure to a low cadmium level was associated with decreased renal function [27]. Another study reported that exposure to high levels of lead and cadmium reduced eGFR and increased the albumin to creatinine ratio, adversely affecting renal function [28]. Furthermore, recent studies have revealed that with an increase in plasma cadmium concentration, the risks of long-term kidney transplant failure and reduced kidney function increase [29]. These findings suggest that exposure to cadmium, lead, and arsenic is associated with CKD. Because the kidney is the main organ responsible for toxin excretion from blood, it is susceptible to the toxicity of heavy metals such as lead, cadmium, and arsenic [30,31]. Cadmium, lead, and arsenic metabolism can produce reactive oxygen species, induce oxidative stress, and cause kidney damage [32,33,34]. Lead exposure promotes lipid peroxidation and the degradation of phospholipids in kidney cells, leading to a loss of cell membrane integrity and nephrotoxicity, or a loss of mitochondrial function in proximal tubular cells [35,36].

A recent clinical trial indicated that a baseline vitamin B_12_ level of ≥248 pmole/L and folate treatment were associated with an increased reduction in the OR of CKD progression [11]. However, another study reported that folate, vitamin B_12_, homocysteine, and cysteine were not related to the CKD stage [13]. The relationship of hyperhomocysteinemia, folate, and vitamin B_12_ with CKD progression is controversial [12]. By contrast, a high level of plasma vitamin B_12_ was related to all-cause mortality after adjustment for renal function and other confounding factors [37,38]. A previous study found elevated plasma vitamin B_12_ concentrations in patients with liver disease, autoimmune disease, and kidney disease [39]. Why the vitamin B_12_ in the plasma of CKD patients is higher than that in the control group is not fully understood. The liver is the largest reservoir of vitamin B_12_ in the body, which may be the destruction of the absorption of vitamin B_12_ by the liver; alternatively, increased hepatocyte turnover/damage may cause more vitamin B_12_ to leak from the liver, resulting in increased levels of vitamin B_12_ in the plasma [34]. In addition, a high level of plasma vitamin B_12_ may be a response to an increased release of vitamin B_12_ stored in the liver, decreased clearance, upregulation of haptocorrin and transcobalamin synthesis, or decreased affinity of vitamin B_12_ to transporters. These conditions usually result in liver damage or CKD [38]. Furthermore, the findings of the present study suggest that the level of plasma vitamin B_12_ is significantly higher in patients with CKD than in controls. Thus, a high plasma vitamin B_12_ level, but not a high folate level, was associated with CKD. These results may have occurred by chance. Thus, at present, our knowledge regarding the association of high plasma vitamin B_12_ with CKD is incomplete.

In the present study, we observed that high levels of blood lead and plasma vitamin B_12_ tended to interact with CKD. This may be because the high levels of blood lead [5] and plasma vitamin B_12_ (Figure 1) significantly decrease eGFR and increase the OR of CKD or the high levels of blood lead and plasma vitamin B_12_ significantly increase the OR of hyperhomocysteinemia [40], which leads to increased oxidative stress and CKD risk [41]. Thus, an increase in the levels of plasma vitamin B_12_, blood lead or cadmium, and total urinary arsenic increase the OR of CKD.

Some limitations of this study must be considered while interpreting the results. This study is cross-sectional in nature. Patients with CKD recruited in this study were prevalent cases; therefore, the causal relationship of plasma folate and vitamin B_12_, blood cadmium and lead, and total urinary arsenic levels with CKD could not be confirmed. We cannot exclude the possibility of the typical reverse causality. Samples were collected only once to evaluate plasma folate and vitamin B_12_, blood cadmium and lead, and total urinary arsenic levels. However, if all patients maintained a stable lifestyle and had homeostatic metabolism, these measurements may be reliable. Moreover, we did not consider the homocysteine level, lipid profile, and supplement use in this study. For future research, it is necessary to determine the role of homocysteine to assess whether plasma vitamin B_12_ and folate concentrations could affect the metabolism of metals, and thus, affect the risk of CKD. Nevertheless, these findings are crucial to understand potential factors associated with CKD.

## 5. Conclusions

The findings from this study suggest that a high concentration of plasma vitamin B_12_ was related to the risk of CKD after adjusting for other covariates. In addition, this research indicates that there was a possible interaction between plasma vitamin B_12_ and blood lead or cadmium, resulting in an increased risk of CKD. However, the mechanism of this association is not fully understood, and further investigation is warranted to advance the understanding of risks associated with CKD.

## Figures and Tables

**Figure 1 nutrients-13-03841-f001:**
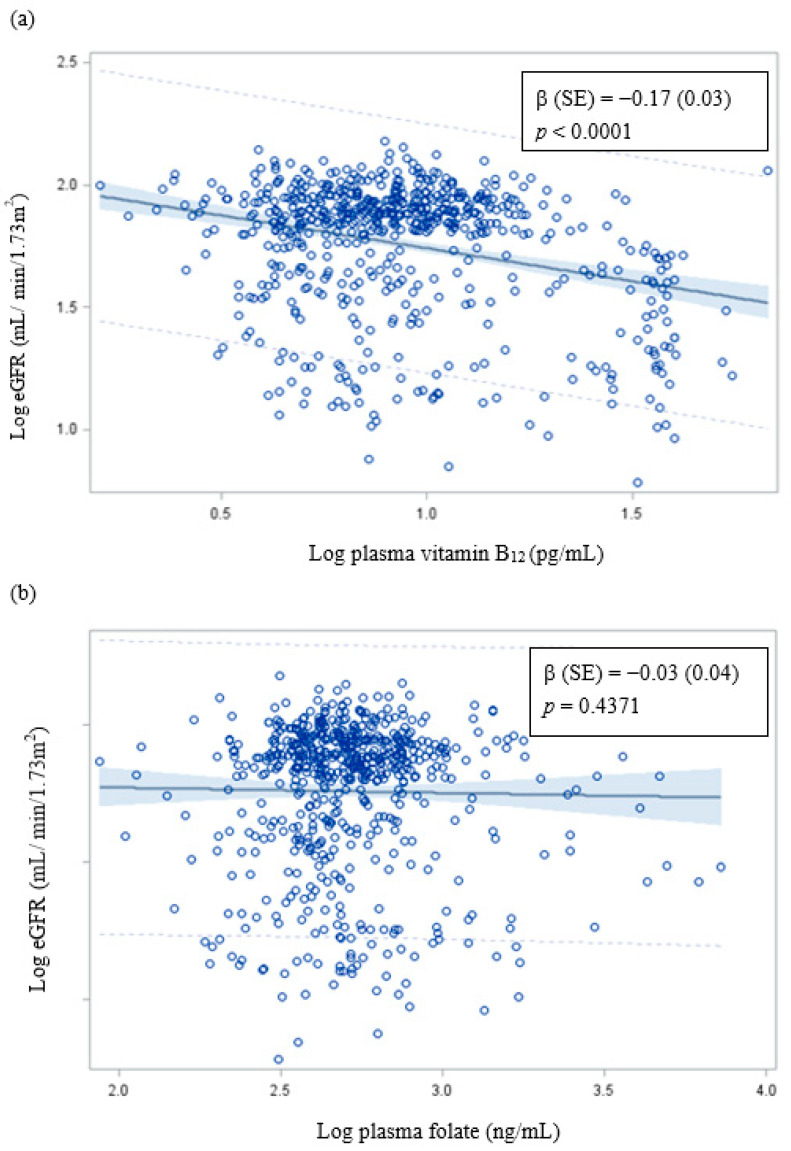
Correlation of eGFR with (**a**) plasma vitamin B_12_ and (**b**) plasma folate. β (SE): Adjusted for age; sex; educational level; alcohol, coffee, and tea consumption; analgesic usage; diabetes; hypertension; blood lead and cadmium levels; and total urinary arsenic (μg/g creatinine).

**Table 1 nutrients-13-03841-t001:** Sociodemographic characteristics, lifestyle, and disease histories of CKD cases and controls.

Variables	CKD Cases (*n* = 220)	Controls (*n* = 438)	*p* Value
Age (years)	65.1 ± 13.5 66.0 (19.0)	64.2 ± 12.5 65.0 (18.0)	0.3796
Sex			
Male	135 (61.4%)	270 (61.6%)	0.9444
Female	85 (38.6%)	168 (38.4%)	
eGFR (mL/min/1.73 m^2^)	31.6 ± 14.6 32.2 (25.2)	84.3 ± 15.7 81.0 (19.3)	<0.0001
Educational level			
Illiterate/elementary school	92 (41.8%)	100 (22.8%)	<0.0001
Junior/senior high school	72 (32.7%)	152 (34.7%)	
College and above	56 (25.5%)	186 (42.5%)	
Cigarette smoking			
Nonsmoker	162 (73.6%)	319 (72.8%)	0.7197
Former smoker	33 (15.0%)	75 (17.1%)	
Current smoker	25 (11.4%)	44 (10.1%)	
Alcohol consumption			
Never	181 (82.3%)	279 (63.7%)	<0.0001
Occasional or frequently	39 (17.7%)	159 (36.3%)	
Coffee consumption			
Never	171 (77.7%)	225 (51.4%)	
Occasional or frequently	49 (22.3%)	213 (48.6%)	<0.0001
Tea consumption			
Never	124 (56.4%)	157 (35.8%)	<0.0001
Occasional or frequently	96 (43.6%)	281 (64.2%)	
Analgesic use			
No/yes as needed	192 (87.3%)	419 (95.7%)	<0.0001
Yes, routinely	28 (12.7%)	19 (4.3%)	
Diabetes			
No	134 (60.9%)	393 (89.7%)	<0.0001
Yes	86 (39.1%)	45 (10.3%)	
Hypertension			
No	96 (43.6%)	306 (69.9%)	<0.0001
Yes	124 (56.4%)	132 (30.1%)	

Values expressed as the mean ±  standard deviation, or median (IQR) for age and eGFR or the number (percent).

**Table 2 nutrients-13-03841-t002:** Association of the levels of total urinary arsenic, blood cadmium and lead, and plasma vitamin B_12_ and folate with CKD.

Variables	CKD Cases (*n* = 220)	Controls (*n* = 438)	Age–Sex Adjusted OR (95% CI)	Multivariate Adjusted OR (95% CI)
Total urinary arsenic (μg/g creatinine)	27.3 ± 21.7 22.5 (18.8)	19.9 ± 13.8 ^#^ 16.0 (15.9)		
≤12.07	36 (16.4%)	146 (33.3%)	1.00 ^§^	1.00 ^§,a^
>12.07–21.90	70 (31.8%)	146 (33.3%)	1.95 (1.23–3.11) **	1.80 (0.98–3.31)
>21.90	114 (51.8%)	146 (33.3%)	3.22 (2.06–5.05) **	2.65 (1.45–4.82) **
Red blood cell lead (μg/L)	69.0 ± 38.9 63.7 (46.6)	41.8 ± 22.8 ^#^ 37.4 (27.0)		
≤27.94	19 (8.6%)	146 (33.3%)	1.00 ^§^	1.00 ^§,b^
>27.94–46.36	46 (20.9%)	136 (31.1%)	2.65 (1.47–4.77) **	2.56 (1.20–5.45) *
>46.36	155 (70.5%)	156 (35.6%)	7.87 (4.61–13.44) **	4.92 (2.42–9.99) **
Red blood cell cadmium (μg/L)	2.4 ± 3.5 1.7 (1.5)	1.2 ± 0.9 ^#^ 1.0 (0.8)		
≤0.80	20 (9.1%)	149 (34.0%)	1.00 ^§^	1.00 ^§,b^
>0.80–1.30	47 (21.4%)	147 (33.6%)	2.57 (1.44–4.60) **	2.30 (1.07–4.98) *
>1.30	153 (69.6%)	142 (32.4%)	8.77 (5.13–14.98) **	6.48 (3.02–13.90) **
Plasma vitamin B_12_ (pg/mL)	15.6 ± 13.2 8.6 (21.9)	8.7 ± 5.0 ^#^ 7.8 (5.1)		
≤6.27	68 (30.9%)	158 (36.1%)	1.0 ^§^	1.0 ^§,c^
>6.27–9.54	52 (23.6%)	140 (32.0%)	0.87 (0.56–1.34)	0.87 (0.48–1.57)
>9.54	100 (45.5%)	140 (32.0%)	1.66 (1.12–2.45) *	2.02 (1.15–3.55) *
Plasma folate (ng/mL)	701.6 ± 856.0 465.0 (339.0)	590.7 ± 454.4 503.0 (270.0)		
≤422	89 (40.5%)	157 (35.8%)	1.00	1.00 ^c^
>422–589	60 (27.3%)	142 (32.4%)	0.74 (0.50–1.11)	1.02 (0.58–1.80)
>589	71 (32.3%)	139 (31.7%)	0.89 (0.60–1.32)	0.99 (0.57–1.72)

Values are expressed as the mean ± standard deviation, median (IQR) for total urinary arsenic, red blood cell lead and cadmium, and plasma vitamin B_12_ and folate or the number (percent). * *p* < 0.05, ** *p* < 0.01, ^#^
*p* < 0.05 for the Wilcoxon rank sum test, ^§^
*p* < 0.05 for the trend test. ^a^ Adjusted for sex; age; educational level; alcohol, coffee, and tea consumption; analgesic usage; diabetes; hypertension; red blood cell lead and cadmium levels; and plasma vitamin B_12_ level. ^b^ Adjusted for sex; age; educational level; alcohol, coffee, and tea consumption; analgesic usage; diabetes; hypertension; urinary creatinine; total urinary arsenic (μg/L); and levels of red blood cell lead or cadmium and plasma vitamin B_12_. ^c^ Adjusted for sex; age; educational level; alcohol, coffee, and tea consumption; analgesic usage; diabetes; hypertension; urinary creatinine; total urinary arsenic (μg/L); and red blood cell lead and cadmium.

**Table 3 nutrients-13-03841-t003:** The interaction between plasma vitamin B_12_, urinary arsenic, and red blood cell lead and cadmium levels on CKD.

Variables	Variables	Case/Control	Age–Sex Adjusted ORs (95% CI)	Multivariate Adjusted ORs (95% CI)
Plasma vitamin B_12_ (pg/mL)	Urinary arsenic (μg/g creatinine)			
≤7.76	<16.01	27/116	1.00 ^§^	1.00 ^§,a^
>7.76	<16.01	31/103	1.33 (0.74–2.39)	1.49 (0.71–3.15)
≤7.76	≥16.01	71/112	2.77 (1.65–4.66) **	2.13 (1.08–4.18) *
>7.76	≥16.01	91/107	3.81 (2.26–6.42) **	4.09 (2.04–8.21) **
	Synergistic index		1.34 (0.64–2.81)	1.91 (0.64–5.64)
	*p* _interaction_		0.3886	0.7213
Plasma vitamin B_12_ (pg/mL)	Red blood cell lead (μg/L)			
≤7.76	<37.37	19/108	1.00 ^§^	1.00 ^§,b^
>7.76	<37.37	25/111	1.32 (0.68–2.54)	1.53 (0.68–3.40)
≤7.76	≥37.37	79/120	3.84 (2.17–6.80) **	3.18 (1.54–6.57) **
>7.76	≥37.37	97/99	5.84 (3.28–10.41) **	5.26 (2.51–11.00) **
	Synergistic index		1.53 (0.78–3.02)	1.57 (0.61–4.06)
	*p* _interaction_		0.9892	0.8834
Plasma vitamin B_12_ (pg/mL)	Red blood cell cadmium (μg/L)			
≤7.76	<1.02	19/106	1.00 ^§^	1.00 ^§,b^
>7.76	<1.02	24/110	1.30 (0.67–2.52)	1.74 (0.76–4.02)
≤7.76	≥1.02	79/122	3.90 (2.20–6.92) **	2.76 (1.32–5.78) **
>7.76	≥1.02	98/100	6.40 (3.54–11.56) **	4.68 (2.18–10.04) **
	Synergistic index		1.69 (0.85–3.35)	1.46 (0.55–3.89)
	*p* _interaction_		0.3599	0.5206

^a^ Adjusted for sex; age; educational level; alcohol, coffee, and tea consumption; analgesic use; diabetes; hypertension; and red blood cell lead and cadmium levels. ^b^ Adjusted for sex; age; educational level; alcohol, coffee, and tea consumption; analgesic use; diabetes; hypertension; and levels of urinary arsenic (μg/g creatinine) and red blood cell lead or cadmium. * *p* < 0.05, ** *p* < 0.01, and ^§^
*p* < 0.05 for the trend test; *p* _interaction_: *p* value for multiplicative interaction.

## Data Availability

The data that support the findings of this study are available on reasonable request from the corresponding author Hsi-Hsien Chen 570713@yahoo.com.

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
