# Peer review of "Plasma Vitamin B12 and Folate Alter the Association of Blood Lead and Cadmium and Total Urinary Arsenic Levels with Chronic Kidney Disease in a Taiwanese Population"

_nutrients, 2021, doi:10.3390/nu13113841_

Round 1

Reviewer 1 Report

This is an interesting study that examined the OR whether the levels of  vitamin B12 and folate alter the OR of CKD incidence.

Some points that should be taken into consideration before publication:

1) Lines 49-50: In Taiwan, the CKD prevalence was 11.9% [3] and the prevalence of end-stage renal disease was the highest in the world [4]; 

Ref #3 was from 2008 and ref #4 from 2016. Please indicate the time that data were obtained. 

2) Lines 61-62: Exposure to arsenic, lead, and cadmium appears to be related to CKD occurrence. 

Please provide the appropriate reference 

3)Materials and methods: Please provide clear the inclusion and exclusion criteria of the participants: age, health status, race etc

4) Line 110: Continuous variables are presented as the mean ± standard error. 

Why SE was used instead of Standard Deviation (SD)? Please also provide SD. You can keep SE as supplemental information.

4) Line 165,167,169: "consumption" Please correct the font size 

Author Response

Dear Reviewer 1,

Thank you for your review. We have responded to your comments point-by-point.

Sincerely,

Hsi-Hsien Chen, MD, PhD,

Division of Nephrology, Department of Internal Medicine, Taipei Medical University Hospital, and Division of Nephrology, Department of Internal Medicine, School of Medicine, College of Medicine, Taipei Medical University, Taipei, Taiwan.

Reviewer 2 Report

serum determination of B12, folate, lead and cadmium should be presented with appropriate details. 

when data is presented as mean, why group differences is tested using non-parametric Wilcoxon rank sum test. the authors should present median and IQR

table 1 repeats  top portion once again at the bottom.

the authors should describe main characteristics of the study population in the results section.

table 2 has formatting issues. pl fix them

how was the cutoff values for table 2 were determined, it apprears that tertiles was used, but it is not mentioned anywhere in the statistical analysis section or mentioned in the result section. what type of relationship was studied and presented as table 2 is not clear. 

multivariate model presented in table 2 adjusted for Vit B12 but not for folate, the reason is not clear.

authors should present the dotplot for the variables f measured to show the spread of data.

Why did authors not measured homocysteine, the amino acid is closely tied to B12 and folate metabolism and in some cases involved clearance of heavy metals.

based on table 2, discussion on higher levels of B12 in CKD patients is not very convincing

purpose of para 2 of discussion is not at all clear.

the limitation of the study is cross-sectional nature.

the authors should discuss their results in light of published studies, not present literature review.

conclusions should be rewritten.

why is "References" present in conclusions. 

Author Response

Dear Reviewer 2,

Thank you for your review. We have responded to your comments point-by-point.

Sincerely,

 Hsi-Hsien Chen, MD, PhD,

Division of Nephrology, Department of Internal Medicine, Taipei Medical University Hospital, and Division of Nephrology, Department of Internal Medicine, School of Medicine, College of Medicine, Taipei Medical University, Taipei, Taiwan.

Round 2

Reviewer 2 Report

two minor comments:

  1. Add “continuous variables are presented as mean+SD or median (IQR) “ in statistical analysis section.
  2. Identify variables for which median (IQR) is presented in table 1.
